# Tracking multiple components of a nuclear wavepacket in photoexcited Cu(I)-phenanthroline complex using ultrafast X-ray spectroscopy

Tetsuo Katayama [1,2], Thomas Northey[3], Wojciech Gawelda [4,5], Christopher J. Milne [6], György Vankó [7], Frederico A. Lima[4], Rok Bohinc [6], Zoltán Németh[7], Shunsuke Nozawa[8,9], Tokushi Sato[4,10], Dmitry Khakhulin[4], Jakub Szlachetko [11], Tadashi Togashi[1,2], Shigeki Owada[1,2], Shin-ichi Adachi[8,9], Christian Bressler[4,12], Makina Yabashi [2] & Thomas J. Penfold[3]

Disentangling the strong interplay between electronic and nuclear degrees of freedom is essential to achieve a full understanding of excited state processes during ultrafast non-adiabatic chemical reactions. However, the complexity of multi-dimensional potential energy surfaces means that this remains challenging. The energy flow during vibrational and electronic relaxation processes can be explored with structural sensitivity by probing a nuclear wavepacket using femtosecond time-resolved X-ray Absorption Near Edge Structure (TR-XANES). However, it remains unknown to what level of detail vibrational motions are observable in this X-ray technique. Herein we track the wavepacket dynamics of a proto-typical [Cu(2,9-dimethyl-1,10-phenanthroline)$_2$]$^+$ complex using TR-XANES. We demonstrate that sensitivity to individual wavepacket components can be modulated by the probe energy and that the bond length change associated with molecular breathing mode can be tracked with a sub-Angstrom resolution beyond optical-domain observables. Importantly, our results reveal how state-of-the-art TR-XANES provides deeper insights of ultrafast non-adiabatic chemical reactions.

[1] Japan Synchrotron Radiation Research Institute, Kouto 1-1-1, Sayo, Hyogo 679-5198, Japan. [2] RIKEN SPring-8 Center, 1-1-1 Kouto, Sayo, Hyogo 679-5148, Japan. [3] Chemistry-School of Natural and Environmental Sciences, Newcastle University, Newcastle Upon-Tyne NE1 7RU, UK. [4] European XFEL, Holzkoppel 4, 22869 Schenefeld, Germany. [5] Faculty of Physics, Adam Mickiewicz University, 61-614 Poznań, Poland. [6] SwissFEL, Paul Scherrer Institut, 5232 Villigen-PSI, Switzerland. [7] Wigner Research Centre for Physics, Hungarian Academy of Sciences, 1525 Budapest, Hungary. [8] Institute of Materials Structure Science, High Energy Accelerator Research Organization (KEK), 1-1 Oho, Tsukuba, Ibaraki 305-0801, Japan. [9] Department of Materials Structure Science, School of High Energy Accelerator Science, The Graduate University for Advanced Studies, 1-1 Oho, Tsukuba, Ibaraki 305-0801, Japan. [10] Center for Free-Electron Laser Science, Deutsches Elektronen-Synchrotron DESY, Notkestrasse 85, 22607 Hamburg, Germany. [11] Institute of Nuclear Physics, Polish Academy of Sciences, 31-342 Kraków, Poland. [12] Centre for Ultrafast Imaging CUI, University of Hamburg, 22761 Hamburg, Germany. Correspondence and requests for materials should be addressed to T.K. (email: tetsuo@spring8.or.jp) or to T.J.P. (email: tom.penfold@ncl.ac.uk)

Charge transfer (CT) states are observed across photo-chemistry and photobiology. Their characteristics are essential in many light-activated processes in nature and for developing novel architectures for solar energy conversion and storage, photocatalysis, and light-emitting devices. However, owing to the often strongly non-Born-Oppenheimer behavior of the excited CT states, full understanding of their properties can only be achieved when studied synchronously with the influence of the nuclear motion. Indeed, these motions drive the mixing of electronic and vibrational wavefunctions forming the ladder of vibronic energy levels upon which excited state dynamics evolve, and can often be important for modulating excited state functional properties[1–3].

This coupled vibrational and electronic energy flow can be investigated by probing the motion of a nuclear wavepacket using femtosecond (fs) pump-probe spectroscopies. The vibrational coherence generated by an ultrashort pump pulse propagates along the main pathways before dissipating through various electronic and vibrational degrees of freedom. However, although pump-probe spectroscopies using ultrashort optical laser pulses are able to probe spectral signatures associated with the specific vibrational motions, it is difficult for systems containing more than a few nuclear degrees of freedom to directly relate these changes with intramolecular bond lengths or bond angles because of the lack of atomic structural sensitivity.

Direct structural insight can be achieved using fs time-resolved X-ray Absorption Near Edge Structure (TR-XANES) that provides sensitivity to the local electronic and structural dynamics around the absorbing atom. The advent of X-ray free electron lasers (XFEL)[4–8], which generate coherent X-ray pulses with an unprecedented peak power (>10 GW) and an ultrashort pulse duration (<10 fs), has paved the way for this technique to be pushed into the ultrafast regime and to allow for tracking the nuclear wavepacket in "real-time", meaning the timescale when the coherent molecular vibrations take place. Recently Lemke et al.[9] used TR-XANES to probe the wavepacket dynamics of solvated $[Fe^{II}(bpy)_3]^{2+}$ (bpy = 2,2'-bipyridine). They not only observed a clear signature of the Fe-N breathing mode, i.e. position of the wavepacket, widely reported in the literature but also extracted information about the width of the wavepacket and the subsequent energy dissipation in the vibrationally hot high-spin excited state. However, in this work, the single Fe-N vibrational mode was perceptible among many vibrational modes because of its dominating character on the spectroscopic observable. Consequently, it remains unknown to what extent detailed

information about vibronic motion can be observed in TR-XANES, in particular for reactions involving more subtle structural changes or multiple reaction coordinates, beyond a strongly displacive and symmetric expansion or contraction of metal-ligand bonds as in case of spin transitions[9,10] in octahedral complexes.

This question is especially pertinent in the context of transition metal (TM) complexes, which are ideal candidates for TR-XANES and combine a number of fundamental phenomena, including electron transfer, intersystem crossing (ISC), internal conversion, and intramolecular vibrational redistribution (IVR) during their excited state dynamics. Importantly, the interpretation of these processes is often complicated due to a large number of vibrational degrees of freedom and excited states, which are highly coupled. A prototypical example is the copper(I)-phenanthroline complex, $[Cu(dmphen)_2]^+$ (dmphen = 2,9-dimethyl-1,10-phenanthroline), which has been studied from both experimental and theoretical perspectives[11–27]. From these previous studies, the general picture emerging is that after excitation into the Metal-to-Ligand Charge Transfer (MLCT) states by visible light, the molecule relaxes into the lowest triplet ($T_1$) state with a flattened structure through the pseudo Jahn-Teller (PJT) distortion within ~10 ps (Fig. 1a, b). During this photoexcited decay, Iwamura et al.[16], using ultrafast optical absorption spectroscopy, observed distinct wavepacket oscillations that were damped with the time constant of ~0.8 ps (Fig. 1c). The oscillations were dominated by 125 cm$^{-1}$ and 290 cm$^{-1}$ vibrations and assigned as the breathing and twisting modes of the complex, respectively. Nevertheless, no information about the magnitude of the structural change was obtained.

Herein we report on a fs TR-XANES study of $[Cu(dmphen)_2]^+$ in acetonitrile solution conducted at SPring-8 Angstrom Compact Free Electron Laser (SACLA)[5]. The measured structural dynamics exhibit subtle, but clearly detectable, coherent wavepacket oscillations superimposed on top of a large background associated with the electronic transient signal. The sensitivity to each vibrational mode contributing to the wavepacket strongly depends on the incident X-ray probe energy, i.e., the final state of the transition, and is uncorrelated to the primary electronic transient change. Quantum dynamics simulations are used to support the interpretation of the experimental results and ascertain the structural change and wavepacket dispersion of the molecular ensemble during the first picosecond (ps) timescale of the phototriggered relaxation dynamics.

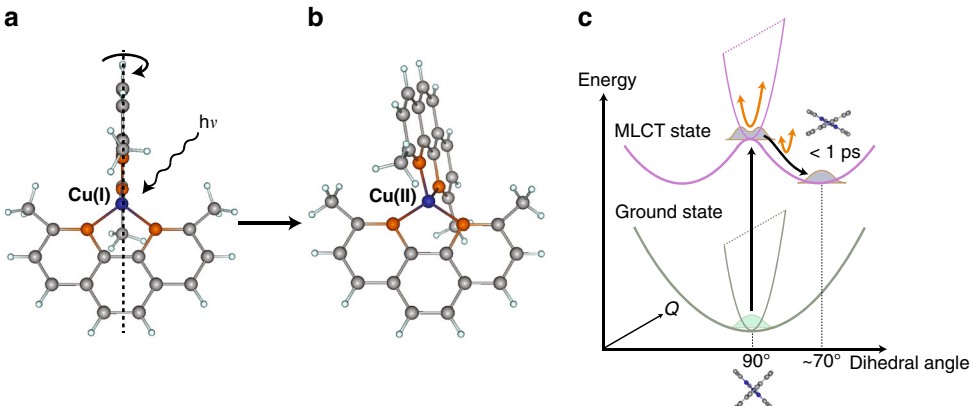

**Fig. 1** Schematics of $[Cu(dmphen)_2]^+$. **a** The structure of the ground $S_0$ state. The symmetry is $D_{2d}$, where two planar dimethyl-phenanthroline ligands coordinate to Cu perpendicularly. **b** The structure after the PJT distortion triggered by photoinduced MLCT excitation. The symmetry is reduced to $D_2$ by the flattening of the dihedral angle between two dimethyl-phenanthroline ligands. **c** Potential energy surface landscape upon which the molecules relax into the PJT distortion

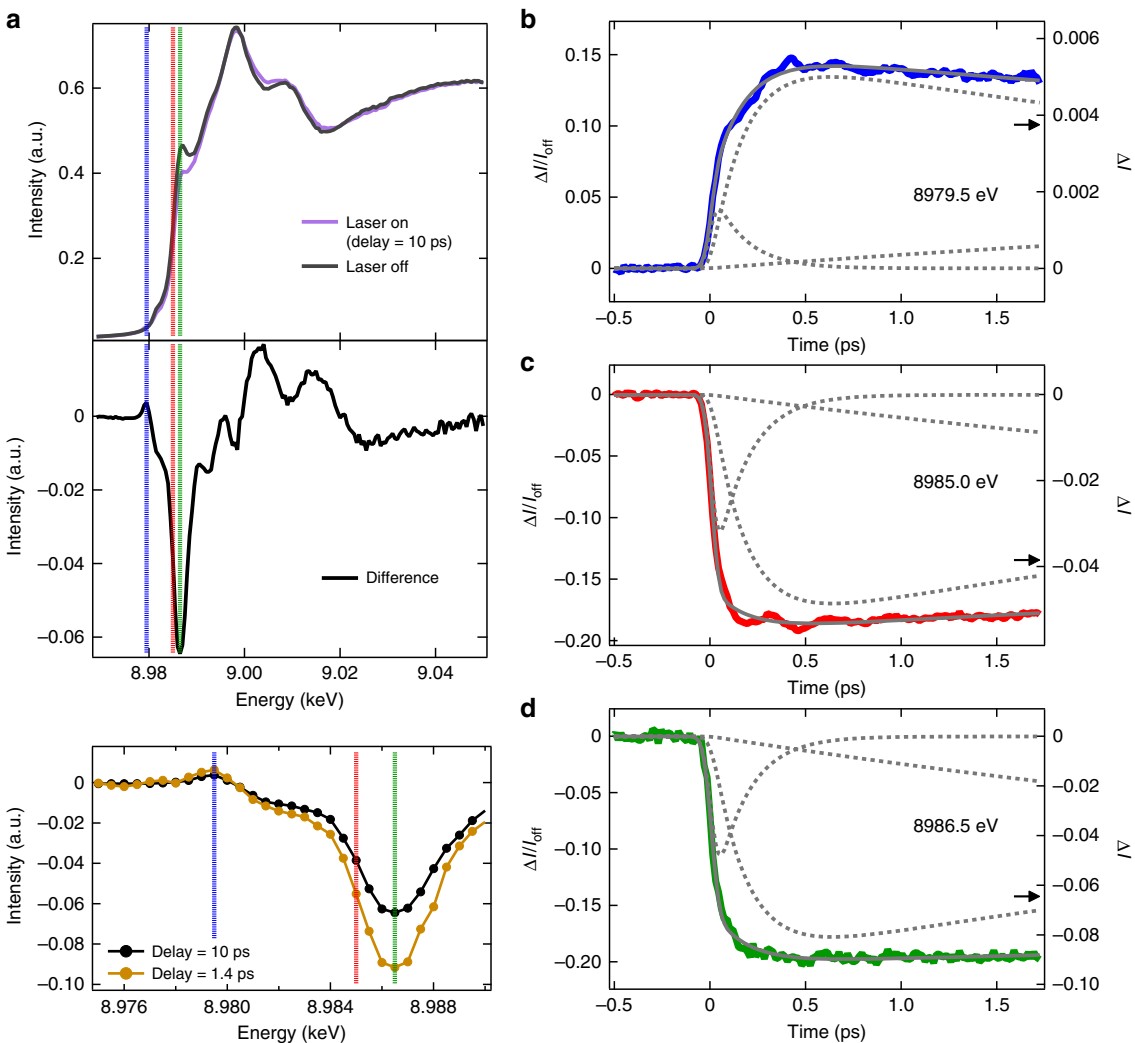

**Fig. 2** Femtosecond dynamics of $[Cu(dmphen)_2]^+$. **a** The top presents Cu K-edge XANES spectra of the $[Cu(dmphen)_2]^+$ ground state (a black line) and the $T_1$ state (a purple line) measured at 10 ps after optical laser irradiation. The middle shows the difference spectrum between these two spectra. The bottom is a zoomed view of the difference spectra measured at 10 ps and at 1.4 ps. The blue, red, green dotted lines denote an excitation photon energy of 8979.5, 8985.0, and 8986.5 eV, respectively. **b–d** The femtosecond time dependence of the transient XANES signal measured at **b** 8979.5 eV, **c** 8985.0 eV, and **d** 8986.5 eV, respectively. The results of the global fitting analysis are overlaid as gray solid lines on the experimental data. Each multiexponential function used in the fitting analysis is shown as gray dot lines. The arrows in **b–d** correspond to the transient signal intensity at 10 ps shown in the middle and bottom of **a**

## Results

**Coherent wavepacket motions measured with femtosecond TR-XANES.** Figure 2a presents the Cu K-edge XANES spectra of $[Cu(dmphen)_2]^+$ in its ground electronic state and 10 ps after photoexcitation at 550 nm, corresponding to the vibronically allowed transition into the lowest energy singlet excited states. A number of changes are observed, which are more clearly visible in the difference spectrum shown at the middle of Fig. 2a. This transient difference spectrum is in very good agreement with the one measured previously using synchrotron radiation at a time delay of 100 ps[11,12,17]. The increase of the pre-edge intensity at 8979.5 eV (a blue dot line in Fig. 2a) corresponds to a $1s \rightarrow 3d$ transition into the hole created in the highest occupied molecular orbital (HOMO) by photoexcitation. This transition contains some $3d$-$4p$ mixing[17] and is sensitive to the structural symmetry of $[Cu(dmphen)_2]^+$. Consequently, the pre-edge intensity can be used as a molecular fingerprint of the symmetry breaking by the PJT distortion, e.g., the peak intensity becomes higher upon the structural flattening as shown in Fig. 1. The negative features at 8985.0 eV (a red dot line in Fig. 2a) and 8986.5 eV (a green dot

line in Fig. 2a) correspond to a spectral region usually associated with a $1s \rightarrow 4p$ transition. As previously demonstrated in ref. [17], the spectral changes in this region of the transient spectrum can be explained by the blue energy shift of the absorption edge that reflects the change of the oxidation state of the central Cu ion from $d^{10}$ to $d^9$ electronic configuration ($Cu^{1+} \rightarrow Cu^{2+}$).

Femtosecond transient XANES changes ($\Delta I/I_{off}$) were measured at these three photon energies and are shown in Fig. 2b–d. The temporal traces at 8979.5 eV and at 8985.0 eV clearly show damped oscillations originating from coherent molecular vibrations. In contrast, no oscillatory features are observed in the temporal trace at 8986.5 eV, which provides the largest transient signal. This observation unambiguously indicates that the selection of the incident X-ray energy is critically important for capturing the nuclear wavepacket. As theoretically predicted by Capano et al. in ref. [23], these results illustrate an important consideration when attempting to disentangle electronic and nuclear changes in TR-XANES. Indeed, the primary transient features (Fig. 2a) derive from the edge-shift arising from generation of the MLCT state, while the nuclear wavepacket

dynamics appear as a small modulation on top of this background signal. Consequently, the energetic positions of the wavepacket signal in the spectrum do not necessarily, as seen here, correspond to features in the transient spectrum dominated by the electronic change. This is confirmed by the lack of wavepacket dynamics at 8986.5 eV, while clear oscillations are seen at 8985.0 eV.

This situation is in stark contrast to the previous study by Kelley and co-workers[27], who investigated two copper(I) diimine complexes but could not capture the signal arising from the wavepacket dynamics in transient spectra. Compared to this previous study, the difference in spectral observables is remarkable and illustrates both the high signal-to-noise ratio and the high time-resolution of the TR-XANES reported here, which allows us to track the weak oscillatory signal of $[Cu(dmphen)_2]^+$. As the size of the structural change accompanied with the PJT distortion is a contraction of the Cu–N bond length by 0.02 Å from the original length in the ground state[25], the present results illustrate the sensitivity of TR-XANES to the Cu–N bond. In MLCT excited states of other TM complexes of Ru[28], Re[29], and Os[30], the changes of the metal-ligand bond lengths have been reported as the order of 0.01 Å, which is similar to the $[Cu(dmphen)_2]^+$ case. This commonality, widely observed in

coordination chemistry of transition metals, is corroborated by the fact[31] that the MLCT states are generally located in the Franck–Condon region where the molecular geometries are relatively close to the ground state structures.

To extract the oscillatory signals observed in the temporal traces, we carried out global fitting analysis using multiexponential functions with three time constants of 0.17 ps, 6.4 ps, and 1.6 ns (gray dot lines in Fig. 2b–d: see Supplementary Note 1 and 7; Supplementary Figs. 2 and 10; Supplementary Tables 2, 3, and 5). These time constants are somewhat different from those determined with fs optical transient absorption spectroscopy by Iwamura and co-workers[16], who reported three kinetic time scales of 0.92 ps, 9.8 ps and a very long-time component, which were attributed to the PJT flattening of the ligands, the ISC transition, and the phosphorescence lifetime of the $T_1$ state, respectively (Supplementary Fig. 3). The reason for this mismatch is that TR-XANES and the optical absorption spectroscopy probe different relaxation processes (described in detail in the following section).

The residuals after the global fitting analysis (Fig. 3a, b: see Supplementary Note 2 and 3; Supplementary Figs. 4 and 5) should retain signals from coherent molecular vibrations. To visualize each mode in a frequency domain, time-dependent

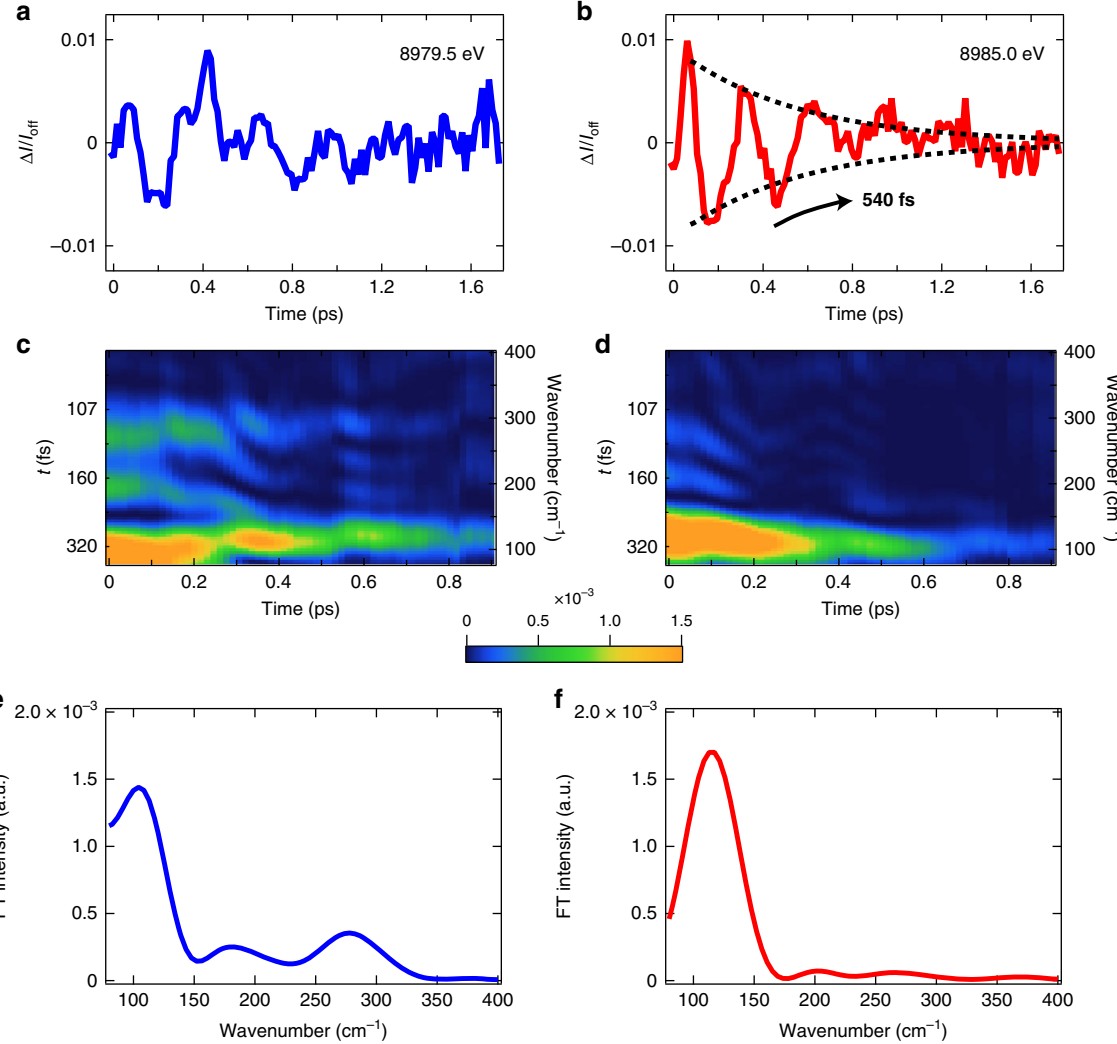

**Fig. 3** Extracted coherent nuclear wavepacket. **a, b** The residual profiles after the global fitting analysis at **a** 8979.5 eV, and **b** 8985.0 eV. At **b** 8985.0 eV, the oscillation decays with the time constant of 540 fs. **c, d** The time-dependent Fourier transform of **a, b** obtained by sliding 1 ps Hann window. The bottom axis is the central time of this Hann window. **e, f** The vertical projection of **c, d** with a time window of 0–0.4 ps

Fourier transforms (FT) of the residuals are also shown in Fig. 3c, d. At the pre-edge (Fig. 3a, c), three distinct bands are found at 83–122 cm$^{-1}$, 165–195 cm$^{-1}$, and 269–287 cm$^{-1}$. In contrast, at the rise of the absorption edge (Fig. 3b, d), only a single band is found at 100–122 cm$^{-1}$. This difference is further validated by the vertical projections of time-dependent FT shown in Fig. 3e, f. Other weak features, above 150 cm$^{-1}$ around ~0.6 ps in Fig. 3c and at 0–0.2 ps in Fig. 3d, have similar intensities to the noise level in a high-frequency region (Supplementary Note 3; Supplementary Fig. 6) and therefore they are not treated as distinct signals. In Fig. 3c, the two bands at 165–195 cm$^{-1}$ and 269–287 cm$^{-1}$ are mostly weakened around ~0.2 ps, while the 83–122 cm$^{-1}$ band is sustained even at 0.5 ps. This behavior indicates that the redistribution of the intramolecular vibrational energy occurs. The possible redistribution mechanism may be the anharmonic vibrational coupling, collisions with solvent molecules, or dephasing. In Fig. 3b, the oscillatory amplitude of the 100–122 cm$^{-1}$ band exhibits an exponential decay with a time constant of 0.54 ps (Supplementary Note 4; Supplementary Fig. 7 and Supplementary Table 4). This decay is accounted for by the loss of the well-defined vibrational coherence while the PJT distortion is proceeding. Although the time constant (0.54 ps) of the PJT distortion in the present fs TR-XANES is shorter than that (0.92 ps) determined by Iwamura et al.[16], the deviation is reasonable if we take into account the different viscosity of solvents used in both studies; The present study uses acetonitrile ($\eta = 0.37$ mPa·s), while Iwamura et al.[16] used dichloromethane ($\eta = 0.44$ mPa s). This explanation is supported by the previous work[15] illustrating that the time constant of PJT distortion can become ~0.4 ps in low-viscosity solvents, agreeing well with our result.

**Quantum dynamics simulations.** To assist the interpretation of TR-XANES, the experimental observable has been simulated using quantum dynamics simulations. The time-dependent simulations are based upon the model spin-vibronic Hamiltonian previously described in refs. [20,21]. Figure 4a shows the populations kinetics of the ground, singlet, and triplet excited states after excitation into the lowest singlet (S$_1$) state using an explicit description of an external electric field with the wavelength of 550 nm. As demonstrated in ref. [20], this shows ultrafast ISC within the first ps. However, we emphasize that owing to the similarity of the singlet and triplet excited state potentials and the lack of direct spin state sensitivity in TR-XANES these states cannot be disentangled in the present experiment.

Figure 4b shows the simulated time-dependent transient spectrum, as described in the methods section below. The structure of the transient spectrum agrees with the experiment (Supplementary Note 5 and Supplementary Fig. 8) and wavepacket oscillations can also be weakly observed. To permit a direct comparison with the experiment, time slices of this transient spectrum are presented with experimental data in Fig. 5. The calculated oscillations show a small phase shift, but are in agreement with the experiment and importantly also exhibit a dependence on the incident X-ray energy. At 8985.0 eV and 8986.5 eV, a period of 330 fs corresponding to the FT 83–122 cm$^{-1}$ as the signal is dominated by the motion of the Cu–N breathing mode ($\nu_8$). The expansion and contraction of metal-ligand bonds associated with this motion modulates the intensity of the absorption edge. Although the calculated oscillations are weaker than observed experimentally, the oscillatory amplitude at 8985.0 eV is larger than that at 8986.5 eV. This behavior is consistent with the experimental observation, which shows the lack of wavepacket motions in the case of the latter. The minimum of the excited state potentials corresponds to the Cu–N contraction of

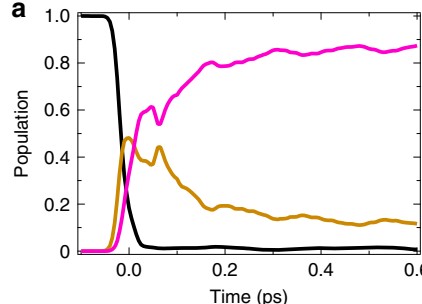

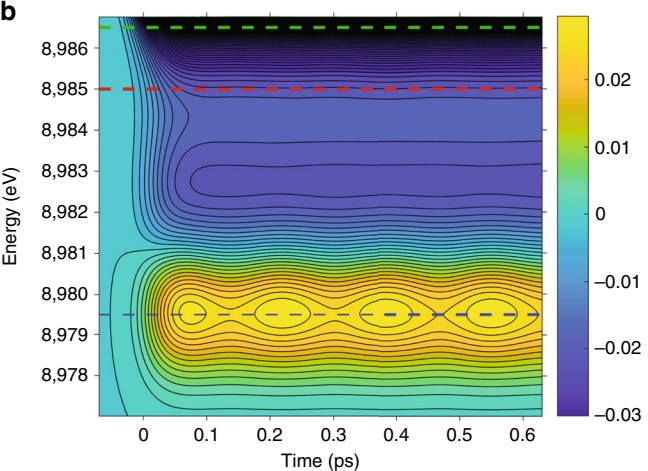

**Fig. 4** Quantum dynamics simulations. **a** Calculated population kinetics of the ground (black), singlet (brown), and triplet (pink) states following the excitation into the S$_1$ state. **b** Time-resolved XANES calculated from this dynamics. The blue, red, and green dashed lines correspond to 8979.5, 8985.0, and 8986.5 eV, respectively

0.02 Å, while the vibrationally hot wavepacket oscillations correspond to a change between +/−0.1 Å from the original Cu–N bond length during the first 0.5 ps. At 8979.5 eV, the oscillation with a period ~170 fs, corresponding to the FT 165–195 cm$^{-1}$ band, is assigned to the normal mode of $\nu_{21}$, which is associated with the PJT distortion as described in refs. [20,21,23]. The appearance of this frequency only at the pre-edge is in good agreement with the experimental data, although in the latter case the $\nu_8$ mode still dominates the measured spectrum (see Fig. 3c). This motion, being related to $\nu_{21}$, breaks the symmetry of the complex and facilitates stronger $3d$–$4p$ mixing, which consequently modulates the intensity of the pre-edge transition. Comparing the vibrational frequencies obtained from the experiment reported here and previous theoretical studies[16,20,21,23], we assigned the FT 269–287 cm$^{-1}$ band in Fig. 3c to the twisting mode ($\nu_{25}$) involved in molecular symmetry breaking.

Owing to the flat nature of the potential energy surface along the PJT mode[16,20] the wavepacket broadens significantly along the $\nu_{21}$ vibration before cooling into the PJT distorted minimum. The dihedral angle between the ligands reduces from 90° to ~70°, but cooling occurs at longer time scales[25] than covered by the present simulations. The evolution of the spectral amplitude from ~500 fs to the spectrum recorded at 10 ps (Fig. 3d) is associated with the wavepacket dispersion along a flat excited state potential and the subsequent vibrational cooling. The latter dynamics cannot be seen in Fig. 5, even if they were to be extended to longer times. This is because the model Hamiltonian contains 8 of a potential 157 normal modes. The effect of this is to confine the

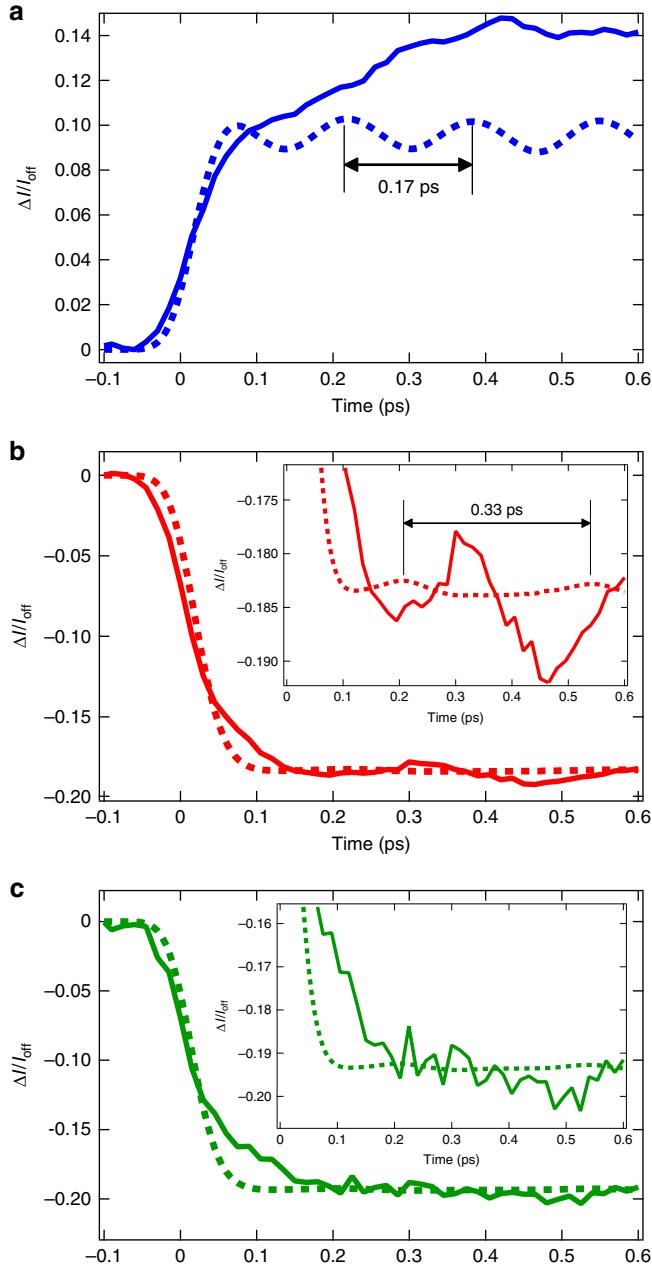

**Fig. 5** Comparison between the experiment data and simulations. Transient spectra at **a** 8979.5 eV, at **b** 8985.0 eV, and at **c** 8986.5 eV as a function of the delay time. Insets of **b**, **c** are zooms into the weak oscillations. Solid line: experiment, dashed line: theory

excess energy generated during the electrical excitation into a smaller subset of modes. This reduces the model's ability to exhibit vibrational cooling, making it over-coherent at longer times (>500 fs) as discussed in ref. [20].

**Discussion of the relaxation time scales**. In the previous section, we have shown the three kinetic components (displayed as dashed lines in Fig. 2b–d) that will be discussed and interpreted in the following. The fastest component of 0.17 ps, not observed in the previous works by Iwamura et al.[15,16], is assigned to the initial Cu–N bond length contraction of the $S_1$ state by 0.02 Å[25] compared to the Franck–Condon structure, which triggers the subsequent dynamics. Indeed, this timescale matches half the

vibrational period of the Cu–N breathing ($\nu_8$, $a_1$) mode, shown as the most intense band in the FT 83–122 cm$^{-1}$. This vibrational mode is most strongly observed on the rising-edge because the contraction (expansion) of the Cu–N bond length leads to a blue (red) shift in the absorption edge[32].

The 0.54 ps component, shown in Fig. 3b, responsible for the loss of vibrational coherence is associated with the PJT distortion[15,16]. This process does not appear as a distinguishable component in the global fitting analysis because the time constant is not well separated from that of the initial Cu–N bond length contraction and the PJT distortion has a weak effect on the transient spectrum[17]. Despite this uncertainty in the global fit, the intensity ratio of the second component with respect to the fastest one is higher at the pre-edge (Fig. 2b) rather than at the rise of the absorption edge (Fig. 2c, d). This trend is consistent with the progress of the PJT distortion as the pre-edge is sensitive to the electronic change accompanied with the symmetry breaking, which modifies the 3d-4p mixing and therefore increases the intensity of the pre-edge. The 6.4 ps time constant is similar to the ~10 ps dynamics assigned to intersystem crossing by Iwamura and co-workers. However, as previously stated, the present experiment is not sensitive to this aspect of the dynamics. Instead, herein we assign this component to vibrational cooling[14]. Finally, the longest 1.6 ns, far beyond the time delays probed in this study, is the lifetime of the long-lived $T_1$ state as previously assigned in ref. [12].

## Discussion

We have demonstrated the sensitivity of TR-XANES, using fs hard X-ray pulses from an XFEL, in tracking multiple components of the coherent nuclear wavepacket dynamics of $[Cu(dmphen)_2]^+$ and disentangled overlapping contributions arising from ultrafast changes in the electronic and geometric structure. Unlike the previous study[9] sensitive to a single dominant reaction coordinate coupled to the symmetric metal-ligand bond elongation by ~0.2 Å[9,33,34], we identify three distinct coherent vibrations whose amplitudes are significantly modified upon varying the incident X-ray energy, i.e. the final state of the transition. The pre-edge intensity primarily stems from the 3d-4p mixing providing some dipole character to the transition, which is enhanced by the symmetry distortion or is mediated by the metal-ligand covalency[35]. Therefore, both the symmetry-breaking modes ($\nu_{21}$ and $\nu_{25}$) and the breathing mode ($\nu_8$) are perceptible at the pre-edge. On the other hand, the rising-edge intensity is sensitive to the shift in the absorption edge reflecting the effective charge on the Cu atom, which is influenced by the average Cu–N bond length. This rationalizes why it is only the Cu–N breathing mode ($\nu_8$), which is observed at the rising-edge. The wavelength-dependent sensitivity is particularly advantageous to investigate the wavepacket dynamics formed by the coherent superposition of multitude of vibrational modes. In addition, our work highlights the unique capability of TR-XANES to gain the detailed information of local structural changes during the wavepacket dynamics. This is well illustrated by the characterization of the Cu–N bond length change associated with the breathing mode ($\nu_8$) at the sub-Angstrom level.

More generally, it is important to identify the electronic changes and nuclear motions playing a key role in an ultrafast chemical reaction. However, not all nuclear motions associated with the wavepacket dynamics are matching or coupled to the reaction coordinate. In the case of $[Cu(dmphen)_2]^+$, the symmetry-breaking modes ($\nu_{21}$ and $\nu_{25}$) are strongly coupled to the ligand flattening motion, while the breathing mode ($\nu_8$) is not as relevant but dominates the wavepacket dynamics. The real-time measurements with a site-specific probe made possible with TR-XANES and the ability to disentangle multiple contributions to the wavepacket demonstrated

herein on the sub-Angstrom level will facilitate a deeper understanding of photocycle of molecules and solid materials.

## Methods

**Experimental set-up.** TR-XANES experiment was performed at BL3[36] of SACLA[5] using a total fluorescence detection method. The XFEL beam was monochromatized by two Si (111) channel-cut crystals with a $(+,-,-,+)$ geometry[37], and was focuesd down to 7 μm by Beryllium compound refractive lenses[37]. [Cu(dmphen)$_2$]PF$_6$ complex was dissolved in acetonitrile with a concentration of 100 mM. The solution was flowed as a liquid jet through an injector with an inner diameter of 50 μm in a closed circulating system. At the interaction point, the sample was excited directly from the ground state into the S$_1$ state by 550 nm optical laser pulses using a Ti:sapphire laser system and optical parametric amplifier (HE-TOPAS, Coherent). The fluence of optical laser pulses was set at 142 mJ cm$^{-2}$ that was slightly lower than the onset of the nonlinear effect (Supplementary Note 6 and Supplementary Fig. 9). The arrival timing between X-ray and optical laser pulses was recorded with the timing diagnostics based on the X-ray beam splitting scheme[38,39]. In the jitter correction, the interval bin width of 15 fs was used. The overall time resolution was evaluated to be ~70 fs in full width at half maximum (FWHM) by a gaussian fit of a first derivative of a time scan at 8986.5 eV, where the largest transient signal was observed in the difference XANES spectrum (the green dot line in Fig. 2a). This instrumental response function (IRF) width is fairly close to a convolution of the XFEL pulse duration (~6 fs), the optical pulse duration (~45 fs), the jitter correction precision (~16 fs), and a group velocity mismatch (~50 fs) between pump and probe pulses in the 50 μm solution (Supplementary Fig. 1 and Supplementary Table 1).

**Computational details.** Quantum dynamics were performed using the multi-configurational time-dependent Hartree (MCTDH) method implemented within the Quantics Quantum dynamics package[40]. These simulations used the model spin-vibronic Hamiltonian previously described in refs. [20,21]. In contrast to previous simulations, which excited the optically bright S$_3$ state[20], in this work, we mimic the 550 nm excitation of the lowest-singlet states by explicating including the laser pulse. The interaction is described using a time-dependent electric field, $\mathbf{E}(t)$, of a pump pulse (with energy of 2.45 eV and a duration of 50 fs with a semi-classical dipole approximation. The description of the transition dipole moment includes a zeroth order term calculated at the Franck–Condon geometry and first-order term along each normal mode coordinate, $Q$ included in the model Hamiltonian to account for vibrational effects on the transition dipole moment.

The time-dependent spectra have been simulated by post-processing quantum dynamics simulations as previously described[23,41]. The X-ray spectrum of the non-stationary wavepacket is calculated as the weighted sum of spectra calculated at each grid point used to describe the nuclear wavepacket. The weighting corresponds to the magnitude of the nuclear wavepacket at that grid point. The ground and excited state spectra were calculated using time-dependent density functional theory (TD-DFT) adapted for the core hole spectra[42] as implemented within the ORCA quantum chemistry package[43]. These calculations were performed within the approximation of the B3LYP exchange and correlation (x-c) potential, using a def2-SVP basis set. The TD-DFT equations were solved for 50 states, within the Tamm-Dancoff approximation and the interaction with the X-ray field was described using the electric dipole + quadrupole approximation. The excited state spectra were all simulated assuming the T$_1$ state using unrestricted Kohn–Sham, due to the insensitivity of XANES to spin structure and details of the electronic structure of the ligands.

## Data availability

All relevant data and programs are available from the corresponding authors upon request.

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

## Acknowledgements

We greatly thank the staff at SACLA for their support and dedicated contributions. This research was performed with the approval of the Japan Synchrotron Radiation Research Institute (JASRI; Proposal No. 2018A8044). T.N. and T.J.P. acknowledge support from the EPSRC (EP/R021503/1) and Leverhulme Trust (RPG- 2016-103). G.V. and Z.N. acknowledge financial support from the "Lendület" (Momentum) Program of the Hungarian Academy of Sciences (LP2013-59), the Government of Hungary and the European Regional Development Fund under grant VEKOP-2.3.2-16-2017-00015, and the National Research, Development and Innovation Fund (NKFIH FK 124460). W.G., F.A.L., D.K., and C.B. received financial support from European XFEL. W.G. further acknowledges partial support from the National Science Center (NCN) in Poland under SONATA BIS 6 grant No. 2016/22/E/ST4/00543, and C.B. acknowledges support from the German Cluster of Excellence CUI: Advanced Imaging of Matter (AIM). JSPS KAKENHI Grant Numbers JP17H06141, JP19H05782, and JP19H04407 supported this work.

## Author contributions

T.K., T.J.P., W.G., C.J.M., G.V., F.A.L., R.B., Z.N., S.N., T.S., D.K., J.S., S.A. and C.B. conceived the proposal and designed the experiment in this project. T.T. and S.O. dedicated for preparation of optical laser and evaluated the optical duration. T.K., W.G., C.J.M., G.V., F.A.L., R.B. and Z.N. performed the experiment and analyzed the measured data. T.J.P. and T.N. performed quantum simulation. T.K., T.J.P., W.G., C.J.M., G.V., F.A.L., C.B. and M.Y. wrote the manuscript with discussion and improvement from all authors.

## Additional information

**Competing interests:** The authors declare no competing interests.

