## [Peer Review File · Nature Communications]

Reviewers' comments:

Reviewer #1 (Remarks to the Author):

This is a report on the femtosecond excited-state dynamics of the prototypical CuI phenanthroline complex $[\text{Cu}(\text{dmphe})_2]^+$ in solution. The authors used time-resolved x-ray absorption spectroscopy (x-ray absorption near edge structure, XANES) at the Cu K absorption edge at the Spring-8 Angstrom Compact Free Electron Laser (SACLA) and quantum dynamics simulations including calculated spectra. The major claim of the study is the observation of hitherto undetected nuclear wavepacket dynamics and a related refinement of the corresponding ultrafast relaxation processes in the photochemical reaction of this complex.

Ultrafast x-ray spectroscopy has matured in the past tens of years to one of the methods of choice for the investigation of photochemical reactions of metal complexes. One reason is the ability to probe structural and electronic structural changes with elemental specificity. X-ray free-electron lasers with unprecedented brilliance and femtosecond pulse duration now offer the opportunity to further advance the method to render finer details of photochemical reactions of metal complexes. This cannot be overrated because time-resolved x-ray measurements at x-ray free electron lasers can contribute to a mechanistic understanding of the femtosecond excited-state dynamics of metal complexes in ways that have been inaccessible before. The present manuscript by Katayama et al. is an excellent example for this. It combines powerful and state-of-the-art experimental and theoretical methods and it reports comparably small but essential effects (compared to earlier reports on other metal complexes).

The observed wavepacket dynamics is made visible by an excellent temporal resolution of the time-resolved pump-probe experiment of less than 100 fs and very good signal to noise ratios. To my knowledge and as the authors also report in their manuscript, this is the third ever report on femtosecond wavepacket dynamics in a metal complex with an x-ray method (after Lemke et al., ref. 9, and Biasin et al., ref. 10 in the manuscript). Impressively, the study reveals three Cu-ligand vibrational modes with initial bond length changes from well below to around 0.1 Angstrom. This is a considerable advancement compared to Lemke et al. and points to using time-resolved XANES to study the complex multidimensional potential energy landscape of metal complexes without restrictions imposed by the method. Further, the predictive character of the employed calculations has to be mentioned as this can be an important guide to experiment and this study is an excellent example for this as well.

The novel information reported by Katayama et al. I think will be of great interest to a large community of scientists using ultrafast x-ray methods in general and spectroscopy in particular

because probing nuclear wavepacket motion in the initial excited-state dynamics of metal complexes is indispensable for establishing reaction coordinates and finding the pathways in the potential energy landscape of the systems. In addition, the study has the potential to appeal to the even wider community of scientists studying the ultrafast excited-state dynamics of metal complexes with non-x-ray methods because it details information that is missing from or cannot be acquired with more established all-optical methods such as time-resolved UV/Vis spectroscopy. The reported combination of methods shows how to reveal the full excited-state dynamics with structural and electronic structural sensitivity including nuclear wavepacket dynamics in photo-excited metal complexes. With this, I believe, the study will influence thinking in the field as it allows for designing new experiments with the aim to uncover hitherto inaccessible information.

The manuscript in my opinion is excellently and clearly written, the science is sound and the presented material seems technically valid.

I want to raise criticism and questions, however, concerning 1) presentation of the results where the authors potentially oversell one aspect of their study, 2) validation of the experimental and theoretical results, 3) illustration of the discussion, 4) the conclusions and 5) details. If these concerns can be removed and questions can be answered by the authors (and some may be due to my misunderstanding), I believe that the manuscript can be improved and the presentation will be more convincing. A correspondingly revised version of the manuscript can then, I think, be an exemplary case with high impact for the application of femtosecond time-resolved x-ray spectroscopy to the photochemistry of metal complexes.

1) Presentation of the results where the authors potentially oversell one aspect of their study

The authors repeatedly claim that they find it important that (quoting here from the abstract) "the coherent vibrational motion associated with the wavepacket dynamics is spectroscopically uncorrelated to the primary electronic change in the transient spectrum". I have to say that I do not quite understand what the authors mean. With some uncertainty due to this lack of understanding I think that they oversell this aspect because I do not consider this very important. Yes, one can claim that the spectroscopic signature of the wavepacket motion is masked by the changes in the spectrum due to changes in the electronic structure. And yes I agree that it is by no means clear a priori whether and if where in the spectrum wavepacket motion will be expressed. However, in the end it is all about the energy potential landscape and transitions between ground and valence excited states and the core-excited final state in the XANES process (for both cases: where the system changes from one to another valence excited state and where it relaxes vibrationally in a given electronic state). In both cases structure and electronic structure are intertwined and affect the spectrum. I just have trouble seeing the distinguishing aspect that leads the authors to their claim (the claim "sounds bigger than it may be"). In any case I think that this claim is not needed because their findings are very interesting also without this aspect.

2) Validation of the experimental and theoretical results

The validation of the results could be strengthened by addressing the following points.

Experiment (on p. 7, last paragraph, and p 8, first paragraph): It would be good to give the uncertainties of the reported frequency values for the three vibrations. In Figure 3c and 3d: Please comment on the faint structures around 0.6 ps at above 150 cm⁻¹ (3c) and 0-0.2 ps at around 200 and 280 cm⁻¹ (3d). They seem weak (below 0.5x10⁻³?) but a short discussion on why the authors consider them statistically irrelevant would be good to enhance validity of those features that are statistically significant. In the Fourier transform analysis leading to the data in Figure 3, what motivates using 1 ps for the size of the Hann window? How robust are the reported frequencies if other window sizes were used? I am confused by the sentence "This behaviour indicates that the redistribution of the intramolecular vibrational energy occurs via the dephasing." I am unsure about what the authors mean. Via what else could energy be redistributed (other than dephasing if that is meant)? Is this the right way of putting it anyway? Isn't it rather that redistribution entails dephasing? Is the 540 fs time scale the result of a fit of the experimental data? If yes, can the authors detail how this fit was done? In the comparison of the present to Iwamura's results, can the authors give the viscosities of the different solvents to get more concrete about how the observed differences relate to these?

Theory (on p. 8, last paragraph, and p. 9, first paragraph): Something seems wrong in the sentence "In the edge region, at... mode (v_8)." I think it will be good to add to this sentence/part of the discussion an explanation for how this vibrational mode relates to changes in orbital mixing and hence spectral changes, similar to the discussion on the following page of the 191 cm⁻¹ band. Related to the claim "the wavepacket broadens significantly along this vibration before cooling into the PJT distortion..." I wonder: This seems to be expressed in the experimental data (damping of the oscillation at 191 cm⁻¹, Figure 3a/c) but not in the calculation (blue line in Figure 4b), see also both in Figure 5a. What is missing in the calculation, can this be briefly discussed? I'm confused about the discussion of the twisting mode (282 cm⁻¹). This doesn't seem to show up in the calculations, why not, what is missing, can this be discussed?

3) Illustration of the discussion

The study would be greatly improved if the authors included (in Fig. 1 e.g.) a schematic of the energy potential landscape (ideally including pathways and time scales). On p. 8 "The structure of the transient spectrum agrees with the experiment...": I find that this claim is very hard to assess and suggest adding a direct comparison of experiment and theory. In Fig. 2 it would be useful if the

reader could relate changes in panel a (relative intensities of the various features in a lower panel) to the changes in panels b-d (relative intensities there, 0 to maximum amplitude e.g.). For this it would be useful to add a transient XANES spectrum at a delay of 1.5 ps in panel 1 (or alternatively show delay scans in b-d all the way to 10 ps).

4) Conclusions

I think that the authors should consider rewriting their conclusions. First paragraph of conclusions: I believe that it is indispensable adding what new information the study provides on the studied Cu complex. Concerning the sentence starting with "The modes that break the molecular symmetry..." I think that the quality of this section would be greatly enhanced if the authors added a very brief description on how distances/symmetry, molecular-orbital mixing and spectral changes relate. I understand that this will lengthen this section but at the same time it would bring everything together at the right place in the manuscript. Second paragraph of conclusions: I find this part quite weak. I don't understand the essence and importance of "spectroscopically uncorrelated" nuclear motion and electronic-structure changes (see 1)). I really do not understand "there is no reason to expect key features in the TR-XANES spectrum recorded on longer (> 100 ps) timescales using synchrotron X-ray pulses to be informative on the fs time regime." Why should ps measurement be informative on fs time scales anyway? The discussion of predicting an experiment with theory rather seems an argument or conclusion for ref. 23 but not for the present study. Can the author instead elaborate on how important it is to detect and characterize the initial nuclear wavepacket dynamics for establishing the reaction coordinates? Other general arguments could be valuable as well.

5) Details

I find the sentence "This mixing is therefore crucial...of functional material." in the introduction very confusing. How can mixing be an approach for enhancing a property? I find the treatment of the results in ref. 10 inappropriate. It is less important than ref. 9 and its discussion should therefore be shorter but I think it is important to better relate to it here. In the Method section (p. 12): How do the authors know that the fluence of the optical pump laser pulses was "slightly lower than the onset of the nonlinear effects"? Is this based on experimental evidence or just an estimate? It would be good to indicate in the caption of Fig. 4 that these (a) are calculated results.

Reviewer #2 (Remarks to the Author):

The manuscript "Tracking multiple components of a nuclear wavepacket in photoexcited Cu(I) phenanthroline complex using ultrafast X-ray spectroscopy" submitted by Tetsuo Katayama and co-workers presents a technically robust experimental and theoretical investigation of the electronic excited state dynamics of the MLCT state of $[\text{Cu}(\text{dmphe})_2]^+$. I see many merits to the study and see how it makes an important contribution to the development of ultrafast hard x-ray spectroscopy, but I also have multiple questions and concerns that need to be addressed to make the discussion more convincing.

(1) A clearer discussion of how electronic and nuclear structural changes appear in the XANES spectrum is needed. For instance, the theoretical work included in the study of Kelley et al. (reference 27) indicate that the structure of the absorption edge depends on both MLCT formation (leading to a shift in the 1s-to-4p resonance) and the flattening of the dihedral angle (leading to a reduction in the 1s-to-4p absorption resonance). This would conform with my expectation that the x-ray absorption edge depends on structure via shape resonance effects. The authors need to convincingly explain why the XANES spectrum is sensitivity to the dihedral angle.

2) I have significant concerns about the wavepacket analysis used at the pre-edge (8979.5 eV). The signal shown in figure 3(a) is the residual from a multi-exponential decay and rise fit to the measured data, with the dominant residual appearing at early times where the signal need not have exponential time dependence (nuclear dynamics need not only appear oscillatory or exponential). The residuals at longer time delays barely exceed the uncertainty of the measurement. While, the evidence for the $\sim 110 \text{ cm}^{-1}$ vibrations is marginal, it appears statistically valid, the evidence supporting the 191 cm^{-1} and 282 cm^{-1} vibrations is dubious, at best, given the uncertainty in the measurement and the uncertainty in the model for the dominant signal seen at 8979.5 eV that dictates the residuals being analyzed.

3) In the comparison between theory and experiment, the measured oscillation frequency ($\sim 110 \text{ cm}^{-1}$) differs significantly from the calculated oscillation frequency ($\sim 150 \text{ cm}^{-1}$, note a period of 225 fs does not have a $\sim 110 \text{ cm}^{-1}$ frequency as stated in the manuscript). This lack of agreement must be understood before the theory can be utilized to explain the structural origin of the 110 cm^{-1} vibration. As discussed in the manuscript, Tahara and co-workers observe a 125 cm^{-1} oscillation that they assign to metal-ligand breathing, a motion that XANES should be able to detect. I did not see where a comparison between the 125 cm^{-1} mode in the Tahara studies is compared with the $\sim 110 \text{ cm}^{-1}$ mode observed in this study.

4) In figure 5(b) and 5(c), the theory predicts a delayed onset in the difference signal not observed in the experiment. Given that charge transfer is effectively instantaneous in the measurement, what is the origin of this delay in difference signal not seen in the measurement?

5) The emphasis on the 'stark contrast' between the work of Lemke et al. (reference 9) and this study is neither convincing nor needed. After all, the 20-30 degree change in the dihedral angle between the two dmphen ligands is far from a small structural change. Understanding the photochemistry and photophysics of copper diimine complexes have merit independent of studies of iron complexes. For this reason, I do not see the work of Lemke as the key study for comparison, but rather the study of Kelley et al. (reference 27).

Lastly, to warrant publication in Nature Communications a more robust description of the significance of the work is needed. At present, the authors appear to argue that the goal of the work is to demonstrate the sensitivity of XANES to a variety of coherent vibrational motions and take the method beyond the work of Lemke et al. The only oscillation that is robustly observed is that ~ 110 cm^{-1} mode which may, in fact, be the metal-ligand breathing mode assigned to a 125 cm^{-1} mode by Tahara (whether the Tahara assignment appears robust and whether or not a 15 cm^{-1} difference in frequency is experimentally robust need to be addressed in the manuscript). If true, I would argue the manuscript is more of a demonstration of the sensitivity of XANES to ultrafast changes in metal-ligand bond lengths (showing sensitivity to bonding changes in hundredths of \AA is significant).

Our responses to the referee's comments are as follows:

To reviewer 1

We thank the referee for careful reading our manuscript and giving useful comments and suggestions, which are much appreciated. In response to referee's comments and advices, we have revised the manuscript. Our point-to-point answers to referee's questions are listed as follows.

Reviewer 1

1) Presentation of the results where the authors potentially oversell one aspect of their study.

The authors repeatedly claim that they find it important that (quoting here from the abstract) 'the coherent vibrational motion associated with the wavepacket dynamics is spectroscopically uncorrelated to the primary electronic change in the transient spectrum';. I have to say that I do not quite understand what the authors mean. With some uncertainty due to this lack of understanding I think that they oversell this aspect because I do not consider this very important. Yes, one can claim that the spectroscopic signature of the wavepacket motion is masked by the changes in the spectrum due to changes in the electronic structure. And yes I agree that it is by no means clear a priori whether and if where in the spectrum wavepacket motion will be expressed. However, in the end it is all about the energy potential landscape and transitions between ground and valence excited states and the core-excited final state in the XANES process (for both cases: where the system changes from one to another valence excited state and where it relaxes vibrationally in a given electronic state). In both cases structure and electronic structure are intertwined and affect the spectrum. I just have trouble seeing the distinguishing aspect that leads the authors to their claim. In any case I think that this claim is not needed because their findings are very interesting also without this aspect.

The statement: 'the coherent vibrational motion associated with the wavepacket dynamics is spectroscopically uncorrelated to the primary electronic change in the transient spectrum' was motivated by the observation that the change that dominates the transient spectrum is the electronic change associated with the generation of the MLCT state (i.e. the edge shift which arises from going from Cu⁺ to Cu²⁺.) To investigate the wavepacket dynamics, it would appear prudent to measure the time-dependence of the largest change in the transient signal occurring at 8986.5 eV. However, the spectroscopic signal associated with the nuclear dynamics (geometry) is not largest here, as shown in the manuscript. Hence the spectral region associated with the electronic change and the nuclear dynamics are uncorrelated.

However, we agree with the referee that this aspect was overemphasised in the original manuscript. It has now been reduced in the present submission.

2) Validation of the experimental and theoretical results

The validation of the results could be strengthened by addressing the following points.

Experiment (on p. 7, last paragraph, and p 8, first paragraph): It would be good to give the uncertainties of the reported frequency values for the three vibrations.

In order to strengthen and support experimental results, we analyzed how the observed peaks shift with respect to the central time in time-dependent FT maps. The obtained frequencies including the uncertainties are 83–122 cm⁻¹, 165–195 cm⁻¹, and 269–287 cm⁻¹ at the 8979.5 eV (Figure 3a,c) and 100–122 cm⁻¹ at 8985.0 eV (Figure 3b,d). We revised the manuscript using these values.

In Figure 3c and 3d: Please comment on the faint structures around 0.6 ps at above 150 cm-1 (3c) and 0-0.2 ps at around 200 and 280 cm-1 (3d). They seem weak (below 0.5x10-3?) but a short discussion on why the authors consider them statistically irrelevant would be good to enhance validity of those features that are statistically significant. In the Fourier transform analysis leading to the data in Figure 3, what motivates using 1 ps for the size of the Hann window? How robust are the reported frequencies if other window sizes were used?

To clarify the robustness of the FT analysis, we confirmed that other weak faint features, above 150 cm⁻¹ around ~0.6 ps in Figure 3c and at 0–0.2 ps in Fugure 3d, have similar intensities to the noise (< 2.5 × 10⁻⁴) observed in the high frequency region (> 400 cm⁻¹; Figure S6 of Supplementary Note 3), where the oscillatory signal should be smeared out due to the time resolution (~70 fs). On the other

hand, the FT 165–195 cm^{-1} and 269–287 cm^{-1} bands have intensities of $\sim 5 \times 10^{-4}$ that is statistically higher than the noise, and therefore we treated them as distinct signals. The number of distinct vibrational modes was further validated by the vertical projection of FT maps with the time window of 0–0.4 ps as shown in Figure 3e,f. The time-dependent FT maps using different window sizes were also added in Figure S5 of Supplementary Note 3, which indicated that the main observables were not affected by this parameter. On the basis of these results, we discussed shortly as following:

“This difference is further validated by the vertical projections of time-dependent FT shown in Figure 3e,f. Other weak features, above 150 cm^{-1} around ~ 0.6 ps in Figure 3c and at 0–0.2 ps in Figure 3d, have similar intensities to the noise level in a high frequency region (Supplementary Note 3) and therefore they are not treated as distinct signals.”

The caption of Figure 3 was also added as following.

“(e,f) The vertical projection of (c,d) with a time window of 0–0.4 ps.”

I am confused by the sentence ‘This behaviour indicates that the redistribution of the intramolecular vibrational energy occurs via the dephasing’; I am unsure about what the authors mean. Via what else could energy be redistributed (other than dephasing if that is meant)? Is this the right way of putting it anyway? ‘it rather that redistribution entails dephasing?’

The mechanism of vibrational energy redistribution was not actually addressed in our experiment. Therefore, we described possible mechanisms as following.

“This behaviour indicates that the redistribution of the intramolecular vibrational energy occurs. The possible redistribution mechanism may be the anharmonic vibrational coupling, collisions with solvent molecules, or dephasing.”

Is the 540 fs time scale the result of a fit of the experimental data? If yes, can the authors detail how this fit was done?

The decay time constant of 0.54 ps of Figure 3b was obtained by fitting the residual containing the oscillatory signal. We added this fitting procedure in Supplementary Note 4 and revised the sentence as following:

“In Figure 3b, the oscillatory amplitude of the 100–122 cm^{-1} band exhibits an exponential decay with a time constant of 0.54 ps (Supplementary Note 4).”

In the comparison of the present to Iwamura’s results, can the authors give the viscosities of the different solvents to get more concrete about how the observed differences relate to these?

We added the sentence describing the difference in the viscosity of solvent as following:

“The present study uses acetonitrile ($\eta = 0.37$ mPa·s), while Iwamura et al.¹⁶ used dichloromethane ($\eta = 0.44$ mPa·s).”

3) Theory (on p. 8, last paragraph, and p. 9, first paragraph)

Something seems wrong in the sentence ‘In the edge region,; mode (v_8)’; I think it will be good to add to this sentence/part of the discussion an explanation for how this vibrational mode relates to changes in orbital mixing and hence spectral changes, similar to the discussion on the following page of the 191 cm^{-1} band.

This sentence was confusing and to clarify we have removed the unnecessary ‘In the edge region’. Also, we have added the sentence describing how the breathing motion appears on the absorption edge as following:

“The expansion and contraction of metal-ligand bonds associated with this motion modulates the intensity of the absorption edge.”

Related to the claim that the wavepacket broadens significantly along this vibration before cooling into the PJT distortion; I wonder: This seems to be expressed in the experimental data (damping of the oscillation at 191 cm⁻¹, Figure 3a/c) but not in the calculation (blue line in Figure 4b), see also both in Figure 5a. What is missing in the calculation, can this be briefly discussed?

The wavepacket motion calculated will not exhibit the same damping with the experiments. This is because the model Hamiltonian used contains 8 of a potential 157 normal modes. The effect of this is that excess energy generated during the electrical excitation will be distributed to a smaller subset of modes, reducing the ability for the model to exhibit vibrational cooling. In essence at longer times (>500 fs) the model will be over coherent. To clarify this we have added the following:

“The evolution of the spectral amplitude from ~500 fs to the spectrum recorded at 10 ps (Figure 3d) is associated with an initial wavepacket dispersion along a flat excited state potential and subsequent vibrational cooling. The latter dynamics cannot be seen in Figure 5, even if they were to be extended to longer times. This is because the model Hamiltonian contains 8 of a potential 157 normal modes. The effect of this is to confine the excess energy generated during the electrical excitation into a smaller subset of modes. This reduces the model's ability to exhibit vibrational cooling, making it over-coherent at longer times (>500 fs) as discussed in ref. 20.”

I'm confused about the discussion of the twisting mode (282 cm⁻¹). This doesn't seem to show up in the calculations, why not, what is missing, can this be discussed?

Despite being a comparatively large amplitude motion, the twisting mode does not actually strongly affect the transient spectrum. Indeed, as shown in ref. 17 of the manuscript (see Figures below), the transient signal on the 100 ps timescale in the region between 8978-8990 eV is described by an edge-shift. This is because the relative changes in the atomic distances, especially in relation to the Cu atom, as a result of the twisting mode are not changed as much as one would imagine by the twisting mode. Consequently, its effect on the low energy region of the XANES spectrum is small.

- (a) **Shifted difference spectrum:** The experimental transient of $[\text{Cu}(\text{dmphen})_2]^+$ at the Cu K-edge at 100 ps (black line) recorded in ref. 17 is simulated by the [(ground state spectrum shifted 1 eV higher in energy_ – (ground state spectrum)]. This so-called shifted difference spectrum accounts for how much of the spectrum can be captured by change in oxidation state of the Cu upon generation of the MLCT. For the region 8978-8990 eV is almost 1-1 agreement. The positive feature not captured is due to the hole generated in the HOMO by photoexcitation which cannot be captured by a simple shifted difference.
- (b) The experimental transient spectrum (black line) and the simulated transient calculated using TDDFT (red line). Figure taken from ref. 17 of the manuscript.

4) Illustration of the discussion

The study would be greatly improved if the authors included (in Fig. 1 e.g.) a schematic of the energy potential landscape (ideally including pathways and time scales).

In the present submission, we have added the schematics of potential energy surface landscape of the Cu(I) phenanthroline complex in Figure 1c. Relating to this change, the caption of Figure 1c was added as following.

“(c) Potential energy surface landscape upon which the molecules relax into the PJT distortion.”

On p. 8: The structure of the transient spectrum agrees with the experimental find that this claim is very hard to assess and suggest adding a direct comparison of experiment and theory.

In Supplementary Note 5 of the present submission, we compared the calculated and experimental transient spectra. Although the delay times of these spectra were different (0.5 ps for the calculation and 10 ps for the experiment), we confirmed a good agreement between them.

In Fig. 2 it would be useful if the reader could relate changes in panel a (relative intensities of the various features in a lower panel) to the changes in panels b-d (relative intensities there, 0 to maximum amplitude e.g.). For this it would be useful to add a transient XANES spectrum at a delay of 1.5 ps in panel 1 (or alternatively show delay scans in b-d all the way to 10 ps).

In the present submission, we have added the zoomed view of the different spectra, measured at 10 ps and 1.4 ps delay times, in the bottom of Figure 2a. We also showed the difference intensity axis on the right side of Figure 2b-d and put arrows to indicate the transient signal intensity at 10 ps delay time. The caption of Figure 2 was revised as following.

“(a) The top panel presents Cu K-edge XANES spectra of the $[\text{Cu}(\text{dmphen})_2]^{2+}$ ground state (a black line) and the T_1 state (a purple line) measured at 10 ps after optical laser irradiation. The middle panel shows the difference spectrum between these two spectra. The bottom panel is a zoomed view of the difference spectra measured at 10 ps and at 1.4 ps. A blue, red, green dot lines denote the excitation photon energy of 8979.5 eV, 8985.0 eV, and 8986.5 eV, respectively. (b-d) The femtosecond time dependence of the transient XANES signal measured at (b) 8979.5 eV, (c) 8985.0 eV, and (d) 8986.5 eV, respectively. The results of the global fitting analysis are overlaid as gray solid lines on the experimental data. Each multiexponential function used in the fitting analysis is shown as gray dot lines. The arrows in (b-d) correspond to the transient signal intensity at 10 ps shown in the middle and bottom panels of (a).”

5) Conclusions

I think that the authors should consider rewriting their conclusions. First paragraph of conclusions: I believe that it is indispensable adding what new information the study provides on the studied Cu complex. Concerning the sentence starting with ‘The modes that break the molecular symmetry’; I think that the quality of this section would be greatly enhanced if the authors added a very brief description on how distances/symmetry, molecular-orbital mixing and spectral changes relate. I understand that this will lengthen this section but at the same time it would bring everything together at the right place in the manuscript.

In the first paragraph of the conclusion of the new submission, we emphasised the sub-ångström level characterization of Cu-N bond length change associated with the breathing mode, which is unique capability of TR-XANES beyond the optical domain observables. This point is also written in the abstract. Also, we described the mechanism how the pre-edge and rising-edge intensities are modified with distance/symmetry changes, which explains the wavelength-dependent sensitivity of TR-XANES to vibrational modes.

Second paragraph of conclusions: I find this part quite weak. I don't understand the essence and importance of & ‘spectroscopically uncorrelated’ nuclear motion and electronic-structure changes (see 1)). I really do not understand ‘there is no reason to expect key features in the TR-XANES spectrum recorded on longer (> 100 ps) timescales using synchrotron X-ray pulses to be informative on the fs time regime’ Why should ps measurement be informative on fs time scales anyway? The discussion of predicting an experiment with theory rather seems an argument or conclusion for ref. 23 but not for the present study. Can the author instead elaborate on how important it is to detect and characterize the initial nuclear wavepacket dynamics for establishing the reaction coordinates? Other general arguments could be valuable as well.

In the original manuscript, we emphasised that the signal arising from the nuclear dynamics was uncorrelated to the electronic change associated with the MLCT generation. However, we agree that

this aspect was overemphasised. In the second paragraph of the conclusion of the present submission, we discussed the importance of finding nuclear motions matching or coupled to the reaction coordinate. In the case of Cu complex, the symmetry-breaking modes are coupled to the ligand flattening, while the breathing mode is not so relevant but dominates the wavepacket. Such detailed insights can be achieved by tracking initial nuclear motions associated with the wavepacket dynamics. Our results demonstrate that TR-XANES, a site-specific probe, is useful to establish the relation between initial nuclear motions and the reaction coordinate.

6) Details

I find the sentence 'This mixing is therefore crucial of functional material; in the introduction very confusing. How can mixing be an approach for enhancing a property?

Charge transfer states are used throughout organic electronics, however they often exhibit poor radiative rates due to the poor wavefunction overlap. In addition, intersystem crossing between ^1CT and ^3CT states is formally forbidden as spin-orbit coupling between two states of the same spatial symmetry is 0. However, dynamics in the excited state will often change the energy gap and therefore mixing between states of different character. This mixing can perturb the character of, for example ^1CT and ^3CT , making these forbidden processes allowed. If controlled, it broadens the available properties. The citations and the references therein included at the end of this sentence provide examples of this.

I find the treatment of the results in ref. 10 inappropriate. It is less important than ref. 9 and its discussion should therefore be shorter but I think it is important to better relate to it here.

We agree that ref. 10 is less important than ref. 9 in the manuscript. The ref. 10 is therefore referred as one of octahedral complexes showing the spin transition, in which the metal-ligand bond length is a key structural parameter. In contrast, ref. 9 is very important in the context of the manuscript. We discussed the achievement of ref. 9 using most of the third paragraph in Page 3.

In the Method section (p. 12): How do the authors know that the fluence of the optical pump laser pulses was slightly lower than the onset of the nonlinear effects this based on experimental evidence or just an estimate?

In Supplementary Note 6, we showed the relation between transient signal intensity and the fluence of optical pump laser pulses.

It would be good to indicate in the caption of Fig. 4 that these (a) are calculated results.

The caption of Figure 4 was revised as following.

“(a) Calculated population kinetics of the ground (black), singlet (brown) and triplet (pink) states following the excitation into the S1 state. (b) Time-resolved XANES calculated from this dynamics. The blue, red, and green dashed lines correspond to 8979.5 eV, 8985.0 eV, and 8986.5 eV, respectively.”

Our responses to the referee's comments are as follows:

To reviewer 2

We thank the referee for careful reading our manuscript and giving useful comments and suggestions, which are much appreciated. In response to referee's comments, we have revised the manuscript. Our point-to-point answers to referee's questions are listed as follows.

Reviewer 2

(1) A clearer discussion of how electronic and nuclear structural changes appear in the XANES

spectrum is needed. For instance, the theoretical work included in the study of Kelley et al. (reference 27) indicate that the structure of the absorption edge depends on both MLCT formation (leading to a shift in the 1s-to-4p resonance) and the flattening of the dihedral angle (leading to a reduction in the 1s-to-4p absorption resonance). This would conform with my expectation that the x-ray absorption edge depends on structure via shape resonance effects. The authors need to convincingly explain why the XANES spectrum is sensitivity to the dihedral angle.

The intensity at the 1s-to-4p transition is sensitive to the effective charge on the absorbing Cu atom, which is influenced by the average Cu-N bond length. This explains why the breathing motion contributes to its intensity. However, the flattening of ligands has a very weak effect on this aspect (the change of metal-ligand bond length and a resultant modulation of the Cu effective charge) and therefore the 1s-to-4p transition is spectroscopically insensitive to this motion. Instead, the flattening of ligands appears in the 1s-to-3d transition, where the symmetry distortion enhances 3d-4p mixing and gives some dipole character to this transition. This explains why the symmetry-breaking modes are perceptible only at the pre-edge. We described these mechanisms in the present new submission.

Kelly et al (reference 27) also interpreted their transient spectrum arising from the electronic changes in the same way with the original manuscript. They also explained that the apparent loss of intensity of the Cu 1s → 4p transition arises from a blue shift of the transition caused by a ~3 eV stabilization of the 1s orbital following oxidation of the Cu center. On the other hand, as they claimed, the 1s-to-3d transition obtained most of the oscillator strength from the weak Cu 3d-4p mixing that made it slightly dipole allowed. This interpretation of the transient spectrum, established in Ref 17, is well accepted and convincing for the Cu(I) phenanthroline complex.

2) I have significant concerns about the wavepacket analysis used at the pre-edge (8979.5 eV). The signal shown in figure 3(a) is the residual from a multi-exponential decay and rise fit to the measured data, with the dominant residual appearing at early times where the signal need not have exponential time dependence (nuclear dynamics need not only appear oscillatory or exponential). The residuals at longer time delays barely exceed the uncertainty of the measurement. While, the evidence for the ~110 cm⁻¹ vibrations is marginal, it appears statistically valid, the evidence supporting the 191 cm⁻¹ and 282 cm⁻¹ vibrations is dubious, at best, given the uncertainty in the measurement and the uncertainty in the model for the dominant signal seen at 8979.5 eV that dictates the residuals being analyzed.

Although the FT 165–195 cm⁻¹ and 269–287 cm⁻¹ bands are weak and only appear at early times in FT maps, their intensities (~5 × 10⁻⁴) are statistically higher than the noise (< 2.5 × 10⁻⁴) observed in the high frequency region (> 400 cm⁻¹; Figure S6 of Supplementary Note 3), where the oscillatory signals should be smeared out due to the time resolution (~70 fs) of the experiment. This observation allows us to distinguish these two bands from the uncertainty level of the measurement and to confirm them as distinct oscillatory signals. On the other hand, other weak faint features, above 150 cm⁻¹ around ~0.6 ps in Figure 3c and at 0–0.2 ps in Figure 3d, have similar intensities to the noise (< 2.5 × 10⁻⁴) and therefore we could not treat them as signals. The number of distinct vibrational modes was further validated by the vertical projection of FT maps with the time window of 0–0.4 ps as shown in Figure 3e,f. The time-dependent FT maps using different window sizes were also added in Figure S5 of Supplementary Note 3, which indicated that the main observables were not affected by this parameter. On the basis of these results, we discussed shortly as following:

“This difference is further validated by the vertical projections of time-dependent FT shown in Figure 3e,f. Other weak features, above 150 cm⁻¹ around ~0.6 ps in Figure 3c and at 0–0.2 ps in Figure 3d, have similar intensities to the noise level in a high frequency region (Supplementary Note 3) and therefore they are not treated as distinct signals.”

The caption of Figure 3 was also added as following.

“(e,f) The vertical projection of (c,d) with a time window of 0–0.4 ps.”

3) In the comparison between theory and experiment, the measured oscillation frequency (~110 cm⁻¹) differs significantly from the calculated oscillation frequency (~150 cm⁻¹, note a period of 225 fs does not have a ~110 cm⁻¹ frequency as stated in the manuscript). This lack of agreement must be understood before the theory can be utilized to explain the structural origin of the 110 cm⁻¹ vibration.

As discussed in the manuscript, Tahara and co-workers observe a 125 cm⁻¹ oscillation that they assign to metal-ligand breathing, a motion that XANES should be able to detect. I did not see where a comparison between the 125 cm⁻¹ mode in the Tahara studies is compared with the ~110 cm⁻¹ mode observed in this study.

This is correct, and we thank the reviewer for emphasising this point. In the simulations, the spectra calculated are a weighted (by wavefunction probability) sum of the spectra at each grid point in the quantum dynamics along v_8 (breathing mode) and v_{21} (twisting mode). During reviewing this comment, we identified that the normalisation of the wavefunction had been performed slightly erroneously, and the norm of the sampled grid points was not completely conserved during the simulations. This was responsible for distorting the wavepacket signal associated with the breathing mode to a time scale. In this new submission we have corrected this error and in better agreement with the work of Tahara and that previously published in ref. 23 (manuscript), the ~300 fs oscillations are visible.

We also analyzed how the experimentally observed peaks shift with respect to the central time in time-dependent FT maps. The obtained frequency, including the uncertainties, is described as 83–122 cm⁻¹ at the 8979.5 eV (Figure 3a,c) and 100–122 cm⁻¹ at 8985.0 eV (Figure 3b,d) for the largest oscillatory signal, which agrees well with Tahara's study.

4) In figure 5(b) and 5(c), the theory predicts a delayed onset in the difference signal not observed in the experiment. Given that charge transfer is effectively instantaneous in the measurement, what is the origin of this delay in difference signal not seen in the measurement?

The reviewer is completely correct and indeed, this helped us to identify an error in the wavepacket analysis identified in the response to the previous comment. This delayed onset was a consequence of this error and this has been corrected during this resubmission of the manuscript.

5) The emphasis on the 'stark contrast'; between the work of Lemke et al. (reference 9) and this study is neither convincing nor needed. After all, the 20-30 degree change in the dihedral angle between the two dmphen ligands is far from a small structural change. Understanding the photochemistry and photophysics of copper diimine complexes have merit independent of studies of iron complexes. For this reason, I do not see the work of Lemke as the key study for comparison, but rather the study of Kelley et al. (reference 27).

In the present submission, we described the comparison between our results and ref. 27 as following:

"This situation is in stark contrast to the previous study by Kelly and co-workers²⁷, who investigated two copper(I) diimine complexes but could not capture the signal arising from the wavepacket dynamics in transient spectra. Compared to this previous study, the difference in spectral observables is remarkable and illustrates both the high signal-to-noise ratio and the high time resolution of the TR-XANES reported here, which allows us to track the weak oscillatory signal of [Cu(dmphen)₂]⁺. As the size of the structural change accompanied with the PJT distortion is a contraction of the Cu-N bond length by 0.02 Å from the original length in the ground state²⁵, the present results illustrate the sensitivity of TR-XANES to the Cu-N bond."

Lastly, to warrant publication in Nature Communications a more robust description of the significance of the work is needed. At present, the authors appear to argue that the goal of the work is to demonstrate the sensitivity of XANES to a variety of coherent vibrational motions and take the method beyond the work of Lemke et al. The only oscillation that is robustly observed is that ~110 cm⁻¹ mode which may, in fact, be the metal-ligand breathing mode assigned to a 125 cm⁻¹ mode by Tahara (whether the Tahara assignment appears robust and whether or not a 15 cm⁻¹ difference in frequency is experimentally robust need to be addressed in the manuscript). If true, I would argue the manuscript is more of a demonstration of the sensitivity of XANES to ultrafast changes in metal-ligand bond lengths (showing sensitivity to bonding changes in hundredths of Angstroms; is significant).

First, as discussed above (Q. 2), the number of distinct vibrational modes is different between probe energies at 8979.5 eV and at 8985.0 eV. The bands of 83–122 cm⁻¹, 165–195 cm⁻¹, and 269–287 cm⁻¹ were identified at the 8979.5 eV (Figure 3a,c), while only a single band of 100–122 cm⁻¹ was

observed at 8985.0 eV (Figure 3b,d). In the conclusion of this resubmission, the wavelength-dependent sensitivity of TR-XANES was discussed including the mechanism how the pre-edge and rising-edge intensities are modified with distance/symmetry changes. Also, we emphasised the sub-ångström level characterization of Cu-N bond length change associated with the breathing mode, which is unique capability of TR-XANES beyond the optical domain observables. This point was also written in the abstract. In the second paragraph of the conclusion part, we discussed the importance of finding nuclear motions matching or coupled to the reaction coordinate. In case of the Cu complex, the symmetry-breaking modes are coupled to the ligand flattening, while the breathing mode is not so relevant. Such detailed insights can be achieved by tracking the initial nuclear motions associated with the wavepacket dynamics. Our results demonstrate that TR-XANES, a site-specific probe, is useful to establish the relation between nuclear motions and the reaction coordinate.

Reviewers' comments:

Reviewer #1 (Remarks to the Author):

The authors adequately addressed the concerns I raised based on the originally submitted version of the manuscript. They weakened the originally emphasized and quite generic statements of how experimental observables, wave packet dynamics and electronic changes are correlated and instead focused on how they extracted the weak oscillatory signals and what they mean in the concrete case. It is important that they added information on statistical significance (including absolute sizes of spectral differences for better comparison) and model dependence of the extracted frequencies as well as information on how data were fitted to enhance validity of their results. Importantly, they also enhanced their discussion of how spectral changes reflect structural changes in the molecule and they related these in detail to changes in molecular-orbital mixing. The potential energy curves are useful. They could have been used more often in the discussion throughout the manuscript but that seems a question of style and the discussion I think works well without. The authors improved their conclusions. A more in-depth assessment of the importance of their results for understanding this particular metal complex could have been useful but that seems a question of taste and of where to put the emphasize of the study. Details were adequately treated and the authors added in particular a comparative discussion of the earlier results by Kelley et al (the name in reference 27 is wrong). I can therefore recommend publication of the resubmitted version of this manuscript in Nature Communications.

Reviewer #2 (Remarks to the Author):

While the authors have addresses most of my concerns, they did not satisfactorily address my concern about the analysis of the pre-edge transient signal and the extraction of oscillations. The issue is not just the signal to noise of the measurement compared to the residuals that persist after a kinetic model fit to the pre-edge signal, but the intrinsic limitations of the kinetic model fit. There is no reason to assume the initial signal will have an exponential time dependence appended with damped oscillations. To repeat, the issue is not just the signal to noise, but also the weakness of the underlying assumptions about the temporal dependence of the time resolved signal. Nothing in the response from the authors has changed my view that the Fourier analysis of the residuals measured at 8979.5 eV is unconvincing.

The reason this warrants emphasis is the authors continue to make the observation of multiple oscillation frequencies the critical advance presented in the paper, but they do not convincingly demonstrate this has been achieved in the measurement. This is reflected in their response to my

final comment regarding the significance of the work and the contrast they draw between this work and previous studies they describe in the introduction – “However, in this work, only the single Fe-N vibrational mode was perceptible among many vibrational modes because of its dominating character on the spectroscopic observable. Consequently, it remains unknown to what extent detailed information about vibronic motion can be observed in TR XANES, in particular for reactions involving more subtle structural changes or multiple reaction coordinates.”

Taken in total there is a lot of good work in the manuscript, but the authors chose to emphasize an analysis that is not convincing. I defer to the editor to adjudicate this impasse and determine the appropriateness of the manuscript for publication in Nature Communications.

To Reviewer 2: We thank the referee for their careful reading of our manuscript. In response to referee's concern, we have revised the manuscript and answer the question raised below.

The issue is not just the signal to noise of the measurement compared to the residuals that persist after a kinetic model fit to the pre-edge signal, but the intrinsic limitations of the kinetic model fit. There is no reason to assume the initial signal will have an exponential time dependence appended with damped oscillations. To repeat, the issue is not just the signal to noise, but also the weakness of the underlying assumptions about the temporal dependence of the time resolved signal. Nothing in the response from the authors has changed my view that the Fourier analysis of the residuals measured at 8979.5 eV is unconvincing.

The reason this warrants emphasis is the authors continue to make the observation of multiple oscillation frequencies the critical advance presented in the paper, but they do not convincingly demonstrate this has been achieved in the measurement. This is reflected in their response to my final comment regarding the significance of the work and the contrast they draw between this work and previous studies they describe in the introduction. However, in this work, only the single Fe-N vibrational mode was perceptible among many vibrational modes because of its dominating character on the spectroscopic observable. Consequently, it remains unknown to what extent detailed information about vibronic motion can be observed in TR XANES, in particular for reactions involving more subtle structural changes or multiple reaction coordinates. Taken in total there is a lot of good work in the manuscript, but they authors chose to emphasize an analysis that is not convincing. I defer to the editor to adjudicate this impasse and determine the appropriateness of the manuscript for publication in Nature Communications.

Response: The referee questions the use of the kinetic model presented in the supporting material, but does not detail the specific limitations of the model or propose an alternative approach. Kinetic models, especially those containing an exponential time dependence appended with damped oscillations, are widely and successfully used throughout ultrafast spectroscopy. However, we acknowledge that any kinetic model has its limitations. There are two assumptions made in our analysis to extract wavepacket dynamics, which is achieved by subtracting the global fitting curve from the temporal traces. The first is that the temporal traces are composed of the oscillatory signals appended on the background arising from the electronic changes and they can be disentangled. The second is that the contribution of the electronic changes can be modeled with the superposition of exponential decay functions based on the sequential first-order kinetic model.

The first assumption has been widely used in previous works¹⁻⁵ studying wavepacket oscillations. Crucially this includes the wavepacket oscillations probed in the present molecule $[\text{Cu}(\text{dmphen})_2]^+$ using transient absorption spectroscopy^{2,3}. This assumption is therefore aligned with a lot of previous works. In this approach, the oscillatory components of any traces will only persist in residuals after the fitting, if the fitting curve is smooth and includes no oscillatory function.

For the second assumption, in principle, it is possible to take any other kinetic models. The reason the sequential first-order kinetic model was employed is that this is one of the simplest models and is consistent with other previous studies^{2,6-8}. While simplicity can impose limitations, it also avoids issues with over-fitting by restricting the number of free fit parameters used. The dynamics of the Cu(I) complex might be more complicated than the sequential first-order kinetic model, but this uncertainty will not alter our main claims as far as the fitting curve is smooth and the curve shape resembles the background arising from the electronic changes. In addition, there is presently no clear indication in the published literature to suggest that a more complicated model should be used. To support this, we applied a different kinetic model to temporal traces and showed the results in Supplementary Note 7. This is a simple exponential rise function with only two free parameters (intensity and rise time). Clearly this model is insufficient to reproduce the experimental observables satisfactorily and left offsets in residuals. However, the effect of these offsets mostly appears in a very low frequency region of the time-dependent Fourier Transform maps. As a result, we can confirm three modes at 8979.5 eV and single mode at 8985.0 eV. In general, the complicated model can subtract the background arising from the electronic changes well, but it always involves a risk of over-fitting by using many free parameters. We will not hesitate trying any other models to apply if concerns of Referee #2 can be addressed. Finally, we reiterate that the conclusions obtained from the fit are strongly consistent with the observations made by completely independent quantum dynamics simulations.

As for the comparison of the observed vibrational frequencies in our data and those reported earlier in the literature², we concluded that the agreement is reasonable. The comparison of the experimental Fourier power spectrum reported by Iwamura et al.² and the dominant modes present in our data is very closely matching. The normal modes utilized in the aforementioned work also present some discrepancies compared to vibrational frequencies obtained by them with TD-DFT methods, e.g. Fig. 4 in Ref.². Consani et al. have also reported such disagreement earlier for the case of ultrafast relaxation dynamics in spin transition compounds, where strong vibrational coherences were reported as well⁹⁻¹⁰.

References

1. Lemke, H. *et al.*, *Nat. Commun.* **8**, 15342 (2017)
2. Iwamura, M. *et al.*, *J. Am. Chem. Soc.* **133**, 7728-7736 (2011)
3. Hua, L. *et al.*, *Phys. Chem. Chem. Phys.* **17**, 2067-2077 (2015)
4. Haldrup, K. *et al.*, *Phys. Rev. Lett.* **122**, 063001 (2019)
5. Biasin, E. *et al.*, *Phys. Rev. Lett.* **117**, 013002 (2016)
6. Chen, L. X. *et al.*, *J. Am. Chem. Soc.* **124**, 10861 (2002)
7. Chen, L. X. *et al.*, *J. Am. Chem. Soc.* **125**, 7022 (2003)
8. Iwamura, M. *et al.*, *J. Am. Chem. Soc.* **129**, 5248-5256 (2007)
9. Consani, C., *et al.*, *Angew. Chem. Int. Ed.* **48**, 7184 (2009)
10. Auböck, G., Chergui, M., *Nature Chem.* **7**, 629 (2015)